# Bio-Chemical Fertilizer Improves the Oil Yield, Fatty Acid Compositions, and Macro-Nutrient Contents in *Nigella sativa* L.

**Samira Moradzadeh** [1]**, Sina Siavash Moghaddam** [1,]*****, Amir Rahimi** [1]**, Latifeh Pourakbar** [2] **,**
**Hesham A. El Enshasy** [3,4] **and R. Z. Sayyed** [5]

1   Department of Plant Production and Genetics, Faculty of Agriculture, Urmia University,
    Urmia 5756151818, Iran; s.moradzadehagro95@gmail.com (S.M.); emir10357@gmail.com (A.R.)
2   Department of Biology, Faculty of Science, Urmia University, Urmia 5756151818, Iran;
    l.pourakbar@urmia.ac.ir
3   Institute of Bioproduct Development (IBD), Universiti Teknologi Malaysia (UTM),
    Johor Bahru 81310, Malaysia; henshasy@ibd.utm.my
4   City of Scientific Research and Technology Applications (SRTA), New Bur Al Arab, Alexandria 21934, Egypt
5   Department of Microbiology, PSGVP Mandal's Arts, Science, and Commerce College,
    Shahada 425409, Maharashtra, India; sayyedrz@gmail.com
*   Correspondence: ss.moghaddam@urmia.ac.ir

**Abstract:** The present study evaluated the effects of biofertilizers on qualitative and quantitative traits of *Nigella sativa* L. The experiment was conducted based on a randomized complete block design with 10 treatments and three replications. The treatments included control (no fertilization), U (100% chemical fertilizer as urea at 53.3 kg ha$^{-1}$, Nb (Biofertilizer, Azotobacter vinelandii), Pb (Biofertilizer, Pantoea agglomerans and Pseudomonas putida), Kb (Biofertilizer, Bacillus spp.), NPKb (NPK combined biofertilizers), Nb + 50% U, Pb + 50% U, Kb + 50% U, and NPKb + 50% U. The highest oil percentage was 46.33 percent related to NPK(b) + U50%, and the highest oil yield was 466.43 kg/ha related to NPK(b) + U50%. The highest seed yield (1006.64 kg/ha) was observed in the plants treated with NPK(b) + U50%. The treatment of K(b) + U50% increased myristic acid by 0.35%. The NPK(b) + U50% treatment reduced palmitic, stearic, and linolenic acid by 11.3, 2.01, and 0.31%, respectively. The highest oleic acid (27.85%) and linoleic acid (56.6%) were obtained from the treatment of NPK(b) + U50%, respectively. The plants treated with NPK(b) + U50% showed the highest seed N percentage (%), P content (mg g$^{-1}$) and K content (mg g$^{-1}$). In general, NPK(b) + U50% is the best treatment in line with sustainable agriculture.

**Keywords:** black cumin; fatty acids; seed macro-elements; biofertilizers



## 1. Introduction

Since the side effects of chemical medicines were specified, attention has been drawn to herbal drugs in medical circles. On the other hand, given the possible adverse effects of excessive use of chemical pesticides and fertilizers on the quantity and quality of active ingredients of medicinal plants, many pharmaceutical companies prefer raw materials produced by sustainable and organic systems [1–3]. In addition to their economic value, medicinal plants are also capable of adapting with organic cultivation methods [4]. Thus, their organic production is unlikely to have a negative impact on their medicinal quality [5].

*Nigella sativa* L. (*N. sativa*) is an annual plant in the family Ranunculaceae that grows to the height of 60–70 cm. Its leaves are gray-green with thread-like cuttings, its flowers are colored white to blue, and its fruits are in the form of follicles encompassing numerous black and aromatic seeds. The seeds contain 40% fatty oil and about 1.4% essential oil, and are medicinally used as carminative, cathartic, milk-promoter, anti-constipation, and sexual promoter in men. Although it is a wild species, it is cultivated in some parts of Europe, Eastern Asia, and some regions of Iran [6–8]. Extensive studies on *N. sativa* have been



carried out by various researchers, and a wide spectrum of its pharmacological actions have been found to include hepato-protective, renal protective, gastro-protective and antioxidant properties, etc. Due to its power of healing, *N. sativa* is among the top-ranked evidence-based herbal medicines [8,9].

Fatty acids are classified into two categories, which include saturated fatty acids such as palmitic acid and stearic acid, as well as unsaturated fatty acids such as oleic acid, linoleic acid, and linolenic acid [10,11]. Fatty acids, especially unsaturated fatty acids and essential fatty acids, are used as a qualitative indicator in foods and as indicators determine food nutritional and medicinal properties such as linoleic and linolenic acids. In other words, these essential fatty acids synthesize other polyunsaturated fatty acids through biological pathways. These essential fatty acids can reduce cholesterol and diabetes, as well as improve the condition of patients with various cancers, including breast cancer, which can be done by increasing the body's immune system [12,13].

Seed fatty oil contains fixed oils such as linoleic acid (55.6%), oleic acid (23.4%) and palmitic acid (12.5%). Various studies have shown that black cumin has different compositions of oils and chemical elements, which obviously depend on the variety and climatic conditions and cultivation [14].

Black cumin has been shown to contain linoleic acid, which is effective in improving heart disease and hypertension as well as the prostate. Obviously, black cumin has other compounds such as trans-anethole [8,15].

Some saturated fatty acids, such as palmitic acid, are precursors for making other saturated fatty acids, such as oleic acid [16].

Macro elements in plants play an important role in their quantitative and qualitative performance and can increase growth components and biochemical compounds [17,18].

The biofertilizer *Azotobacter*-1 (containing N-fixing bacteria from the genus *Azotobacter*) is a molecular selective N-fixing compound that is capable of synthesizing active biological compounds such as nicotinic acid, pantothenic acid, biotin, vitamin B group, auxins and gibberellins, etc., in root zones, thereby contributing to the development of root systems. They also influence plant yields and soil attributes by improving the uptake of water and nutrients, and the biological fixation of nitrogen *Azotobacter* can also produce anti-pathogenic compounds and is involved in disease control [19]. The Barvar-2 phosphate biofertilizer contains two phosphate solubilizing bacteria from the species *Bacillus* and *Pseudomonas* that can convert soil phosphorus from an insoluble to absorbable and soluble form by secreting organic acids and acid phosphatase [20]. Likewise, PotaBARVAR-2 is a type of biofertilizer that contains bacteria that release and dissolve potassium in soil solution and thus provide potassium to the plant's root [21].

Biofertilizers are indeed materials containing different free-living microorganisms [22] that can convert macronutrients from unavailable to available forms through biological processes and thereby help the development of root systems and the better germination of seeds [23]. Biofertilizers act through the production of dissolving secretions and the reduction of acidity to make different nutrients available to plants. In addition to fixing atmospheric N and balancing the uptake of macro- and micronutrients required by plants, the bacteria of biofertilizers synthesize and secrete plant growth stimulators, and different amino acids and antibiotics so they contribute to the growth and development of roots and shoot, and this causes the production of more assimilates and their mobilization to other parts of the plants [24]. Studies on the application of inorganic fertilizers, vermicompost, biofertilizers, and compost revealed the positive effect on the growth and yield in *Nigella sativa* [25,26]; however, it appeared to be an economically infeasible approach as it involved a combination of various preparations that increased the cost of treatment. In addition, nitrogen, phosphorus and potassium biofertilizers have not been previously used in combination to reduce the, use of urea as a common chemical fertilizer in the area. The purpose of this study was to determine the effects of a bio-chemical fertilizer system as a practical and cost-effective method on yield components, oil yield, and fatty acid profiles of *Nigella sativa*.

## 2. Materials and Methods

### 2.1. Field Characteristics and the Region Weather Data

The study was carried out in the research farm of the Department of Agriculture, Urmia University, Urmia, Iran. The location of the farm was 45°10′ East, latitude 37°44′ North, and an elevation of 1338 m from sea level.

### 2.2. Soil Preparation and Sowing

Before starting the experiment, samples of farm soil from 0–30 cm depth were taken and analyzed. Then, the farm was leveled and prepared to build 3 × 2.5 m$^2$ plots. Each plot was composed of 12 sowing rows with an inter-row spacing of 25 cm and an on-row spacing of 15 cm. The seeds were sown 11 March 2017, and the plants emerged with rainwater. The first irrigation was carried out two weeks after sowing. The plants were thinned, and earthing up or ridging was conducted around the base of a plant, which keeps the plant strength. The basic weather data during the experiment are shown in Table 1 and the physicochemical characteristics of soil in the study site can be observed in Table 2.

**Table 1.** Basic weather data during the experiment.

| Month | Total Monthly Precipitation (mm) | Average Monthly Temperature (°C) | Average Monthly Relative Humidity (%) |
|---|---|---|---|
| March | 31.3 | 7.2 | 60.1 |
| April | 57.9 | 12.3 | 54.7 |
| May | 42.8 | 16.6 | 57.3 |
| June | 28.7 | 20.6 | 51.4 |
| July | 5.5 | 24.4 | 42.1 |

**Table 2.** The physicochemical characteristics of soil in the study site.

| Measured Trait | Soil of Study Site before Urea Application |
|---|---|
| Sampling depth (cm) | 0–30 |
| Salinity (dS/m) | 1.31 |
| Soil texture | Loam-clay |
| Acidity (pH) | 7.72 |
| Lime % (TNV) | 16.78 |
| Clay (%) | 44 |
| Silt (%) | 35 |
| Sand (%) | 21 |
| Organic carbon (%) | 0.91 |
| Nitrogen (%) | 0.03 |
| Phosphorous (mg/kg) | 10.33 |
| Potassium (mg/kg) | 398 |

### 2.3. Design and Measurements

The study was carried out on the basis of a randomized complete block design with 10 treatments and three replications (30 plots). Seed density as commonly used in West Azerbaijan is 3 kg per hectare.

### 2.4. Treatments

The fertilizer was composed of urea as chemical fertilizer, NPK biofertilizer (Green Biotech Incorporation, Tehran, Iran), and combined urea and biofertilizers. The treatments included control (no fertilization), U (100% chemical fertilizer as urea at 53.3 kg ha$^{-1}$), Nb (Biofertilizer, *Azotobacter vinelandii*), Pb (Biofertilizer, *Pantoea agglomerans* and *Pseudomonas putida*), Kb (Biofertilizer, *Bacillus* spp.), NPKb (NPK, mixed biofertilizer), Nb + 50% U, Pb + 50% U, Kb +50% U, and NPKb + 50% U. A non-fertilized treatment was used as a control for the necessary comparisons. AzotoBarvar-1 contained Azotobacter vinelandii (strain O4), an obligate aerobic free-living Gram-negative soil bacterium that fixes soil N. PhosphoBarvar-2

contained two types of phosphate solubilizing bacteria, *Pantoea agglomerans* (strain P5) and *Pseudomonas putida* (strain P13), that use secretion of organic acids and phosphatase acids for hydrolyzing insoluble P compounds. PotaBarvar-2 included two types of *Bacillus* spp. bacteria solubilizing K. The microorganisms in this fertilizer decompose insoluble K compounds in the soil around rhizosphere and release this cation.

### 2.5. Application of Biofertilizers

All biofertilizers were applied through seed priming before sowing as a recommended method. Initially, seeds were placed in absolute darkness for 6 h in 500 ppm gibberellic acid (as priming treatment) to break dormancy and improve germination. Biofertilizers were applied as seed impregnation, and urea was incorporated with soil.

### 2.6. Measurement of Morphological Traits

Ten samples to determine morphological traits were randomly taken from each plot on July 9 after removing the marginal effect.

### 2.7. Estimation of Fatty Acid Content

Oil was extracted by a Soxhlet extractor and hexane solvent. Fatty acids were determined with a gas chromatography device. Identification of each of the fatty acids was performed using Sigma standard fatty acid mixture by comparing the retention times. (Agilent-6890 Gas Chromatograph Made by Agilent, Santa Clara, CA, USA), was used, as equipped with a capillary injection valve, a 30 m long capillary fatty acid (DB-wax) column and a 0.25 mm internal diameter with 25 mm thick polyethylene glycol stationary phase. The flame ion detector (FID) can approach zero µm.

### 2.8. Nitrogen Content

One gram of ground seeds was mixed with 5 g of catalyzer (copper sulfate, potassium sulfate, and copper oxide), and it was then added with 20 mL of 98% sulfuric acid. The samples were kept at 410 °C in a Kjeldahl (Gerhardt Company, Bonn, Germany) device for 1.5 h [27]. After they were taken out of the device, 20 mL of distilled water was added, and then they were titrated with Titrasol® sulfuric acid. The amount of acid applied in titration was placed in the following equation to yield nitrogen content.

$$\text{Nitrogen}\% = \text{Quantity of acid employed in titration } 0.0014/\text{Sample weight} \times 100 \quad (1)$$

### 2.9. Estimation of Phosphorus Content

To measure *Nigella sativa* phosphorus content, 1 g of the sample was ground and meshed. After it was digested by dry burning (with HCL), it was adjusted to 100 mL by adding distilled water. Then, 5 mL of the sample was mixed with 5 mL of yellow solution (ammonium heptamolybdate + ammonium vanadate) and then its volume was increased to 25 mL by adding distilled water. After 0.5 h, the samples were filtered through a filter paper and the resulting extract was read at 470 nm with a spectrophotometer (Spectrophotometer APEL Model PD303UV, USA). First, phosphorus standards and then the main samples were read. To prepare the standard, 2.19 g of $KH_2PO_4$ was first solved in a slight amount of water and was adjusted in volume in a 1-L volumetric flask (thick standard). For the series of standard solutions, 10, 8, 6, 4, 2, and 0 mL of the thick standard was taken, and 5 mL of zinc molybdate ammonium was added and adjusted to 25 mL.

### 2.10. Estimation of Potassium Content

To determine *N. sativa* potassium content, 1 g of dry ground and meshed sample was placed in a furnace at 550 °C for 24 h. After digestion by dry burning method (with HCL), the samples were adjusted to 100 mL using distilled water. Using a Clinical pfp7 flame photometer (Jenway Gransmore Green Felsted, Dunmow Essex, CM6 3LB, ENGLAND, Model 8515), first potassium standards and then the main samples were read by the flame

emission method. To prepare the standard, 9.53 g of potassium chloride was solved in water and its volume was adjusted in a 1-L volumetric flask (thick standard). Then, for a series of standard solutions, 10, 8, 6, 4, 2, and 0 mL of the thick standard was poured into 100-mL volumetric flasks containing 50 mL of water and 4.5 mL of thick sulfuric acid, and it was adjusted to the desired volume [28].

### 2.11. Statistical Analysis

Data were statistically analyzed by SAS (Ver. 9.4) software package (Statistical Analysis Software, Cary, NC, USA). Means were compared by the PLSD test at the $p < 0.05$ level, and the graphs were drawn with MS-Excel.

## 3. Results

### 3.1. Seed Yield per Hectare

The highest seed yield of 1006.64 kg/ha was obtained from the treatment of $NPK_{(b)}$ + $U_{50\%}$ and the lowest (556.3 kg/ha) was from the control. The treatment of $NPK_{(b)}$ + $U_{50\%}$ had statistically significant differences with other treatments. The plants fertilized with $NPK_{(b)}$ + $U_{50\%}$ and $P_{(b)}$ + $U_{50\%}$ produced 52.42 and 38.18 percent higher seed yield per ha than control (Figure 1A).

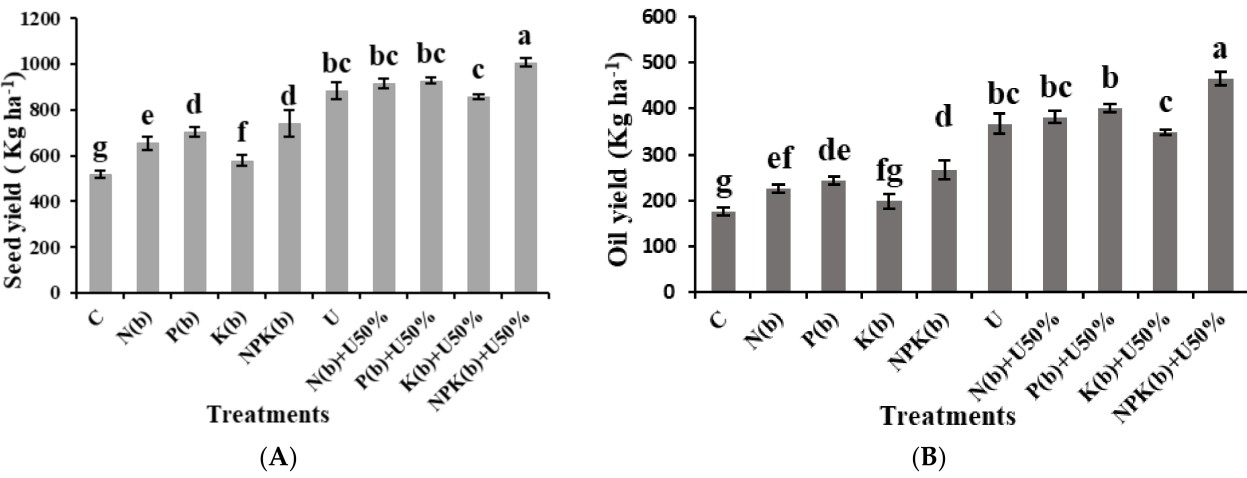

(**A**)　　　　　　　　　　　　　　　　　　　　　　(**B**)

**Figure 1.** Means comparison for seed yield per hectare (**A**) and oil yield per hectare (**B**) of *Nigella sativa* L. as influenced by biofertilizers and chemical urea. Different letters indicate a significant difference according to PLSD test with 3 replications at the $p \leq 0.05$. (chemical urea fertilizer (U), biofertilizer $N_{(b)}$, biofertilizer $P_{(b)}$, biofertilizer $K_{(b)}$, mixed biofertilizer $NPK_{(b)}$, 50% urea + biofertilizer N ($N_{(b)}$ + $U_{50\%}$), 50% urea + biofertilizer P ($P_{(b)}$ + $U_{50\%}$), 50% urea + biofertilizer K ($K_{(b)}$ + $U_{50\%}$), and 50% urea + biofertilizer NPK ($NPK_{(b)}$ + $U_{50\%}$)).

### 3.2. Oil Yield

The highest oil yield (392.47 kg/ha) was obtained from $NPK_{(b)}$ + $U_{50\%}$ and the lowest (190.23 kg/ha) was from control. The treatment of $NPK_{(b)}$ + $U_{50\%}$ was significantly different from other treatments. The plants treated with $NPK_{(b)}$ + $U_{50\%}$ and $P_{(b)}$ + $U_{50\%}$ showed 106.45 and 75.61% higher oil yield than control, respectively (Figure 1B).

### 3.3. Oil Percentage

The highest oil percentage (46.33%) was obtained from $NPK_{(b)}$ + $U_{50\%}$ and the lowest (34.20%) was from the control. The treatment of $NPK_{(b)}$ + $U_{50\%}$ differed from other treatments significantly. This treatment and the treatment of $P_{(b)}$ + $U_{50\%}$ increased oil percentage by 35.47 and 26.16% as compared to the control (Figure 2).

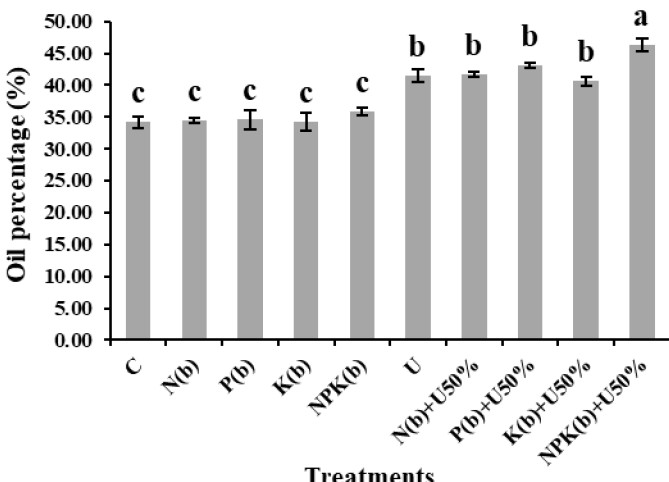

**Figure 2.** Means comparison for seed oil percentage of *Nigella sativa* L. as influenced by biofertilizers and chemical urea. Different letters indicate a significant difference according to PLSD test with 3 replications at the $p \leq 0.05$. (chemical urea fertilizer (U), biofertilizer $N_{(b)}$, biofertilizer $P_{(b)}$, biofertilizer $K_{(b)}$, mixed biofertilizer $NPK_{(b)}$, 50% urea + biofertilizer N ($N_{(b)} + U_{50\%}$), 50% urea + biofertilizer P ($P_{(b)} + U_{50\%}$), 50% urea + biofertilizer K ($K_{(b)} + U_{50\%}$), and 50% urea + biofertilizer NPK ($NPK_{(b)} + U_{50\%}$)).

*3.4. Fatty Acid Profile of Seeds*

3.4.1. Oleic Acid

According to the results about the profile of fatty acids of *N. sativa*, the highest oleic acid percentage (27.85%) was obtained from $NPK_{(b)} + U_{50\%}$ and the lowest (24.43%) was from the control. It was found that $NPK_{(b)} + U_{50\%}$ and $P_{(b)} + U_{50\%}$ increased the oleic acid content by 33.99 and 11.91%, respectively (Figure 3A).

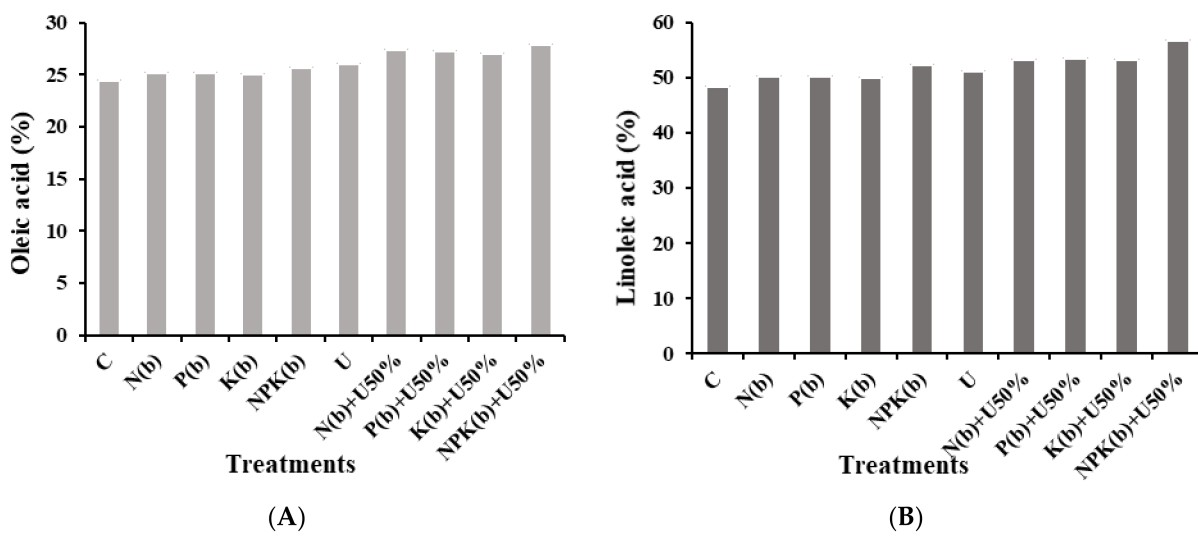

(**A**)                                                                 (**B**)

**Figure 3.** Means comparison for oleic acid (**A**) and linoleic acid (**B**) of *Nigella sativa* L. as influenced by biofertilizers and chemical urea (chemical urea fertilizer (U), biofertilizer $N_{(b)}$, biofertilizer $P_{(b)}$, biofertilizer $K_{(b)}$, mixed biofertilizer $NPK_{(b)}$, 50% urea + biofertilizer N ($N_{(b)} + U_{50\%}$), 50% urea + biofertilizer P ($P_{(b)} + U_{50\%}$), 50% urea + biofertilizer K ($K_{(b)} + U_{50\%}$), and 50% urea + biofertilizer NPK ($NPK_{(b)} + U_{50\%}$)).

### 3.4.2. Linoleic Acid

The highest linoleic acid content of 56.6% was related to $NPK_{(b)} + U_{50\%}$ and the lowest (48.3%) was related to the control. The plants treated with $NPK_{(b)} + U_{50\%}$ and $P_{(b)} + U_{50\%}$ showed 17.18 and 10.55% higher linoleic acid content, respectively (Figure 3B).

### 3.4.3. Linolenic Acid

As the results for the fatty acids profile indicated, the highest percentage of linolenic acid (0.65%) was related to the control and the lowest (0.31%) was related to $NPK_{(b)} + U_{50\%}$. In fact, the application of $NPK_{(b)} + U_{50\%}$ and $P_{(b)} + U_{50\%}$ reduced this fatty acid content by 52.03 and 41.53% versus the control, respectively (Figure 4A).

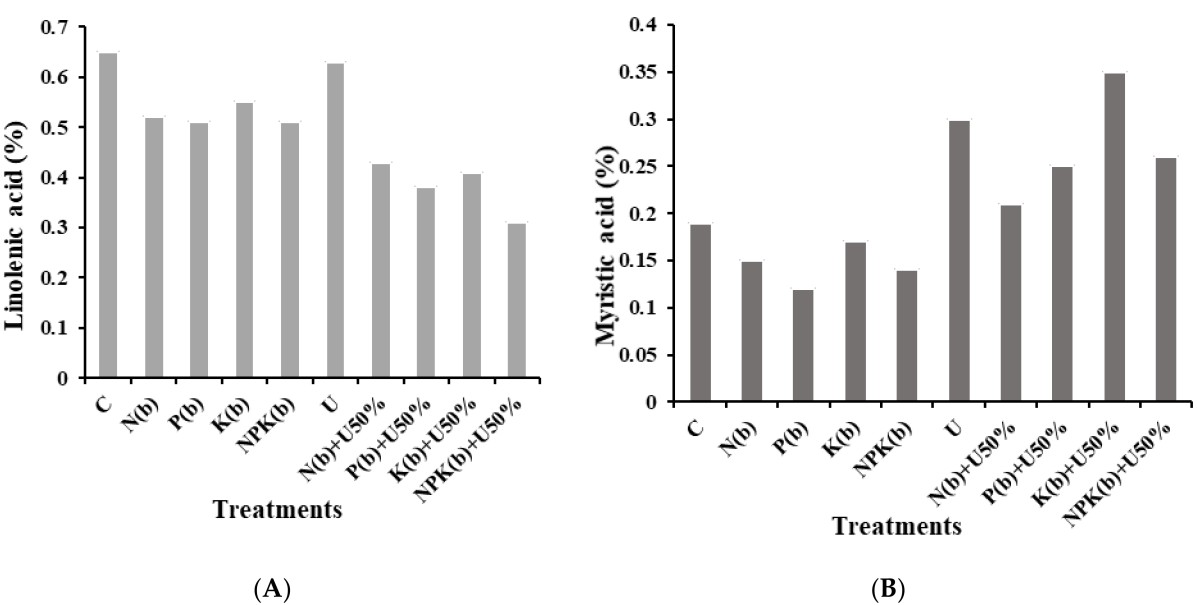

(**A**)　　　　　　　　　　　　　　　　　　　　　　　　(**B**)

**Figure 4.** Means comparison for myristic acid (**A**) and linolenic acid (**B**) of *Nigella sativa* L. as influenced by biofertilizers and chemical urea (chemical urea fertilizer (U), biofertilizer $N_{(b)}$, biofertilizer $P_{(b)}$, biofertilizer $K_{(b)}$, mixed biofertilizer $NPK_{(b)}$, 50% urea + biofertilizer N ($N_{(b)} + U_{50\%}$), 50% urea + biofertilizer P ($P_{(b)} + U_{50\%}$), 50% urea + biofertilizer K ($K_{(b)} + U_{50\%}$), and 50% urea + biofertilizer NPK ($NPK_{(b)} + U_{50\%}$)).

### 3.4.4. Myristic Acid

The plants fertilized with $K_{(b)} + U_{50\%}$ produced the highest myristic acid percentage of 0.35%, and those treated with $P_{(b)}$ produced the lowest (0.12%). The results revealed that the treatment of $K_{(b)} + U_{50\%}$ and U increased the myristic acid content by 191.6 and 150%, respectively (Figure 4B).

### 3.4.5. Palmitic Acid

Palmitic acid maximized to 15.4% in the control plants, but its minimum level was 11.3%, observed in the plants treated with $NPK_{(b)} + U_{50\%}$. It was revealed that the treatments of $NPK_{(b)} + U_{50\%}$ and $N_{(b)} + U_{50\%}$ reduced this acid content by 26.62 and 23.37% versus the control, respectively (Figure 5A).

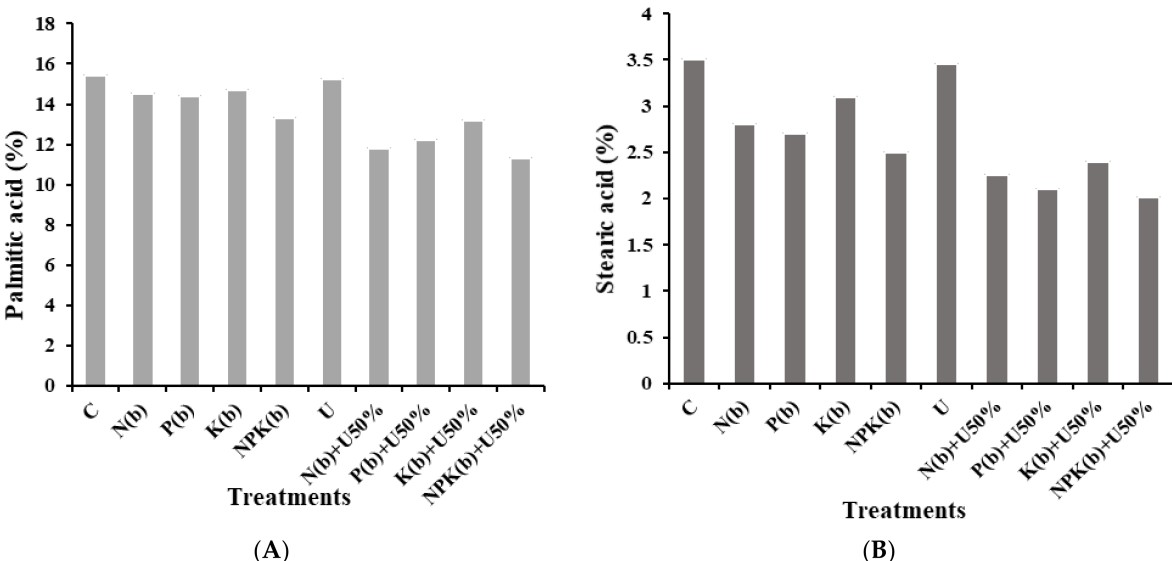

**Figure 5.** Means comparison for palmitic acid (**A**) and stearic acid (**B**) of *Nigella sativa* L. as influenced by biofertilizers and chemical urea (chemical urea fertilizer (U), biofertilizer $N_{(b)}$, biofertilizer $P_{(b)}$, biofertilizer $K_{(b)}$, mixed biofertilizer $NPK_{(b)}$, 50% urea + biofertilizer N ($N_{(b)} + U_{50\%}$), 50% urea + biofertilizer P ($P_{(b)} + U_{50\%}$), 50% urea + biofertilizer K ($K_{(b)} + U_{50\%}$), and 50% urea + biofertilizer NPK ($NPK_{(b)} + U_{50\%}$)).

### 3.4.6. Stearic Acid

It was found that the highest stearic acid content of 3.5% was related to the control and the lowest one (2.01%) was related to the plants treated with $NPK_{(b)} + U_{50\%}$ The plants treated with $NPK_{(b)} + U_{50\%}$ and $P_{(b)} + U_{50\%}$ showed 42.57 and 40% lower stearic acid content than the control, respectively (Figure 5B).

### 3.4.7. Graph of Unsaturated to Saturated Fatty Acid

The ratio of unsaturated fatty acids to saturated fatty acids is shown in the graph (Figure 6), which displays an increase in the ratio of unsaturated fatty acids to saturated fatty acids in NPK (b) + U50%.

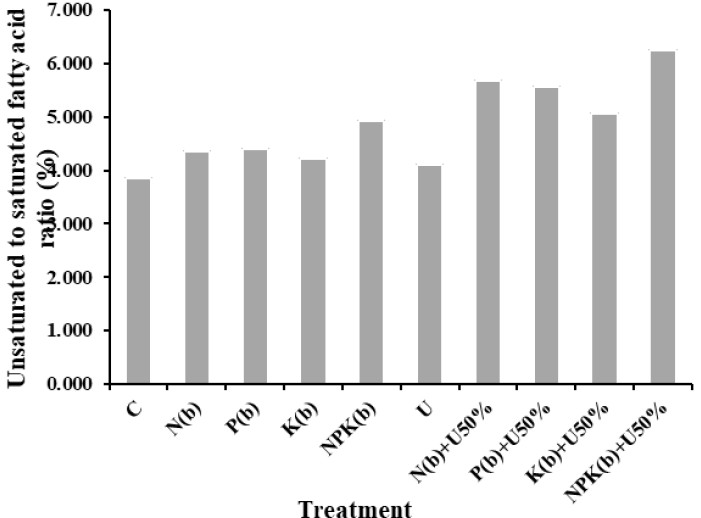

**Figure 6.** Unsaturated to saturated fatty-acid ratio (chemical urea fertilizer (U), biofertilizer N(b), biofertilizer P(b), biofertilizer K(b), mixed biofertilizer NPK(b), 50% urea + biofertilizer N (N(b) + U50%), 50% urea + biofertilizer P (P(b) + U50%), 50% urea + biofertilizer K (K(b) + U50%), and 50% urea + biofertilizer NPK (NPK(b) + U50%)).

### 3.5. The Accumulation of Macronutrients (NPK) in N. sativa Seeds

3.5.1. Seed N Content

The treatment of $NPK_{(b)} + U_{50\%}$ was related to the highest seed N percentage (1.199%), and the control showed the lowest seed N percentage (1.28%). Moreover, the treatment of $N_{(b)} + U_{50\%}$ significantly enhanced seed N percentage higher than that of U treatment. According to the means comparison, the treatments of $NPK_{(b)} + U_{50\%}$ and $N_{(b)} + U_{50\%}$ increased seed N% by 54.80 and 53.24%, respectively (Figure 7).

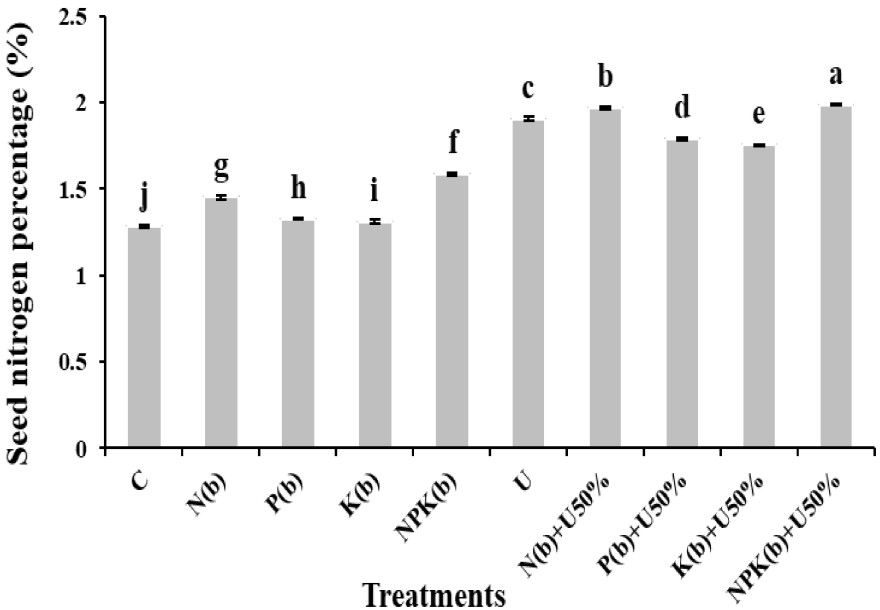

**Figure 7.** Means comparison for seed nitrogen percentage of *Nigella sativa* L. as influenced by biofertilizers and chemical urea. Different letters indicate a significant difference according to PLSD test with 3 replications at the $p \leq 0.05$ (chemical urea fertilizer (U), biofertilizer $N_{(b)}$, biofertilizer $P_{(b)}$, biofertilizer $K_{(b)}$, mixed biofertilizer $NPK_{(b)}$, 50% urea + biofertilizer N ($N_{(b)} + U_{50\%}$), 50% urea + biofertilizer P ($P_{(b)} + U_{50\%}$), 50% urea + biofertilizer K ($K_{(b)} + U_{50\%}$), and 50% urea + biofertilizer NPK ($NPK_{(b)} + U_{50\%}$)).

3.5.2. Seed P Content

The plants treated with $NPK_{(b)} + U_{50\%}$ showed the highest seed P content of 1.30 (mg $g^{-1}$), and the control plants exhibited the lowest one 0.63 (mg $g^{-1}$). The treatment of $NPK_{(b)} + U_{50\%}$ differed from other treatments significantly. According to the means comparison, the treatment of $NPK_{(b)} + U_{50\%}$ increased the seed P content by 105.26%, and the treatment of $P_{(b)} + U_{50\%}$ increased it by 84.21% (Figure 8A).

3.5.3. Seed K Content

The treatment of $NPK_{(b)} + U_{50\%}$ was related to the highest seed K content of 1.92 (mg $g^{-1}$), and the control was related to the lowest one 1.09 (mg $g^{-1}$). The difference of seed K content in plants treated with $NPK_{(b)} + U_{50\%}$ was statistically significant compared to that of the other treatments. According to the means comparison, the treatments of $NPK_{(b)} + U_{50\%}$ and $K_{(b)} + U_{50\%}$ increased the seed K content by 76.68 and 59.20%, respectively (Figure 8B).

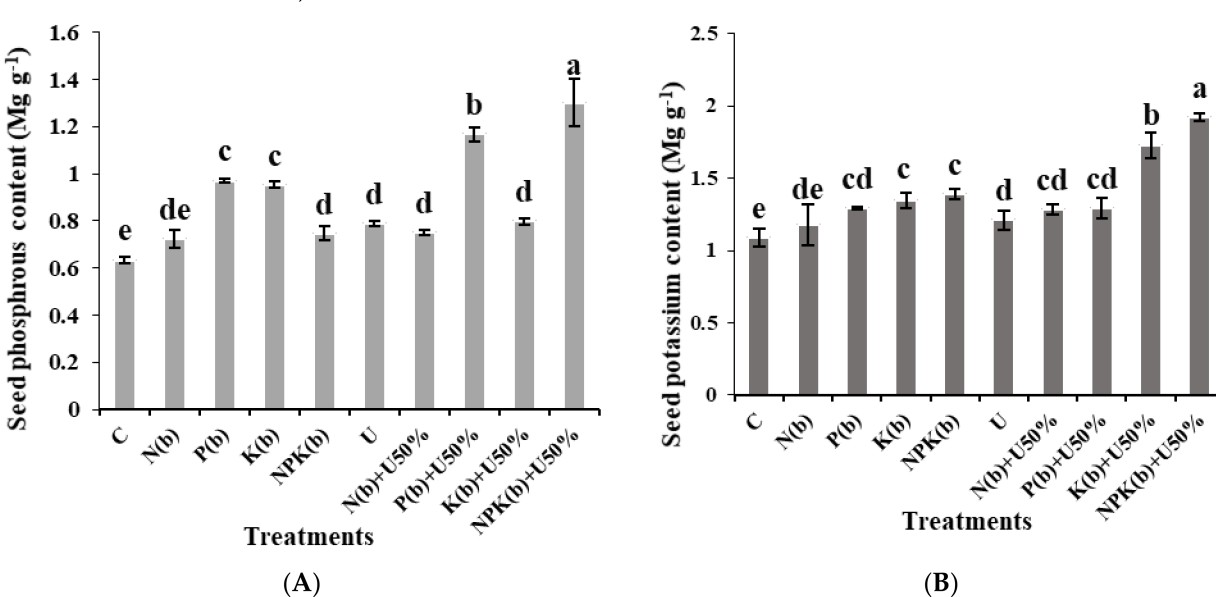

**Figure 8.** Means comparison for seed phosphorous content (**A**) and seed potassium content (**B**) of *Nigella sativa* L. as influenced by biofertilizers and chemical urea. Different letters indicate a significant difference according to PLSD test with 3 replications at the $p \leq 0.05$ (chemical urea fertilizer (U), biofertilizer $N_{(b)}$, biofertilizer $P_{(b)}$, biofertilizer $K_{(b)}$, mixed biofertilizer $NPK_{(b)}$, 50% urea + biofertilizer N ($N_{(b)} + U_{50\%}$), 50% urea + biofertilizer P ($P_{(b)} + U_{50\%}$), 50% urea + biofertilizer K ($K_{(b)} + U_{50\%}$), and 50% urea + biofertilizer NPK ($NPK_{(b)} + U_{50\%}$)).

## 4. Discussion

### 4.1. Seed Yield per Hectare

The results of our study showed that biofertilizers alone or in combination with urea improved seed yield per hectare. There are several reports in connection with the effectiveness of a combination of bio-chemical fertilizers on the quantitative characteristics of plants.

With respect to the increased seed yield in plants fertilized with N-containing biofertilizers, researchers have stated that when plants consume nitrogen and other nutrients, they will be in a better condition for nutrient uptake. The bacteria of biofertilizer contribute to optimal uptake of water and nutrients, and synthesis of growth hormones and some vitamins by their potential for N biofixation and the expansion of root area. Subsequently, this contribution is manifested in higher yield [29]. Similarly, the bacteria of P-containing biofertilizers rely on their potential to make nutrients and organic compounds available to plants to increase their growth and improve the structure and activity of beneficial soil microorganisms, and thereby they grant plants optimal access to water and nutrients, and escalate yield [30]. A study on sunflowers showed that the integrated application of mineral N fertilizer and Azotobacter had a significant effect on improving seed yield and yield components [31]. Dan et al. [32] reported that seed inoculation with *Azotobacter* and Pseudomonas increased seed yield versus control. They attributed this finding to the increase in root length, shoot, and seed yield due to the symbiosis of N-fixing bacteria by supplying N requirement and the existence of an extensive root network.

Research has shown that the combined use of biological and chemical fertilizers, including nitrogen, can increase the quantity and quality of plants, such as oil percentage. The combined use of nitrogen and pseudomonas increased safflower oil [33]. The use of Bacillus also increased plant yield [34].

### 4.2. Oil Yield

The results obtained from this study specify that combined urea–biofertilizer (NPK(b) + U50%) improved oil yield.

The oil of N. sativa has been described as a new and invaluable edible source [35]. It has been reported that environmental conditions and nutrition during growth period of the maternal plant can influence oil content of canola directly [36]. A study has found that applying the appropriate amount of chemical fertilizers [NPK (25:12.5:12.5 kg ha$^{-1}$)] led to the highest oil yield of coriander [37].

Previous research findings indicated that plants treated with chemical urea produced the highest oil yield (1214 kg/ha) [38]. Biofertilizers can increase quantitative and qualitative yields such as grain oil and protein. Experiments have shown that Azoto Barvar-1 and Phosphate Barvar-2, as biological fertilizers, can make these elements (N,P) accessible in the soil [39,40]. The availability of nutrients can promote plant metabolism and carbohydrate transformation, which result in enhancement of plant grain yield and oil content. For this reason, potassium, along with thiobacillus, which are used as biofertilizers, can increase the yield and percentage of fennel oil [41,42].

### 4.3. Oil Percentage

We found that combined urea–biofertilizers (NPK(b) + U50%) enhanced oil percentage compared to the control.

It has been documented that the application of 100 kg/ha N or P can increase the oil content of *N. sativa* seeds significantly [43]. Previous research findings demonstrated that the inoculation of groundnut with biofertilizers increased its oil percentage also [44]. Some researchers have attributed the significant increase in oil content of sunflower seeds to the application of biofertilizer [40]. In a study on the effect of N fertilization on sunflowers, the highest oil percentage (8.46%) was obtained from the application of urea [38]. The application of biofertilizer enhances oil percentage, and this has been attributed to the effects of these fertilizers on increasing photosynthesis through improving water and nutrient uptake, which contributes to producing more assimilates and improving growth, and finally, fatty oil percentage is increased versus uninoculated seeds [45].

### 4.4. Fatty Acid Profile of Seeds

The results of our study showed that combined urea–biofertilizers (NPK$_{(b)}$ + U$_{50\%}$) improved seed fatty acids.

In other reports also, linoleic, oleic, and palmitic acids were identified as the most abundant unsaturated and saturated fatty acids in *N. sativa* oil [35]. Safflower response to chemical fertilizers and biofertilizers is essential for improving its yield and economic profitability. The inoculation of seeds with biofertilizers reduces the content of saturated fatty acids (palmitic and stearic acids) and increases the content of unsaturated fatty acids such as linoleic and organic acids [33]. Researchers conducted experiments on soybeans and rapeseed using fertilizers and achieved similar results to increase the oil content of these plants [36,46]. It has been reported that inoculation with biofertilizer increases the fatty acid content of soybean. Means comparison showed that N application and inoculation increased linoleic acid content [46]. Chloroplasts are the major source of saturated and unsaturated fatty acids in plant cells, and nearly 95% of them are produced by plastidial fatty acids such as oleic acid—stearic acid and linoleic acid [47,48]. It should be noted that unsaturated fatty acids play an important role in the plant defense system, which depends on their carbon bond and arrangement, such as the resistance of tomatoes and eggplants to powdery mildew and *Verticillium dahliae*, diseases, respectively, or avocado and tomato resistance to *Colletotrichum gloeosporioides* and *Pseudomonas syringae* disease as well. Biofertilizers in beans increase fatty-acid content, which in turn makes the plant resistant to *Botrytis cinerea* [48]. The mechanism of production of fatty acids is such that they are produced in plastidial by acetyl coenzyme A, which in turn is produced through the plastid pyruvate dehydrogenase pathway. Another possible route is the activation and import of free acetate through coenzyme A synthetize [49]. In general, two enzymes are involved in the production of fatty acids, acetyl CoA carboxylase and fatty acid synthase, which are present in the chloroplast stroma [49,50]. In accordance with the results of our



research, the work of other studies verified that the combined use of bio-chemical fertilizers and consequently less use of chemical fertilizers increases unsaturated fatty acids such as linoleic acid in the plant, which is one of the most important fatty acids in human health and also makes the plant resistant to adverse environmental conditions [51]. It is now well established that the combined use of fertilizers reduces saturated fatty acids. These results have been proven in soybeans and safflower [33].

### 4.5. The Accumulation of Macronutrients (NPK) in N. sativa Seeds

#### 4.5.1. Seed N Percentage

We found that interaction of urea—biofertilizer ($NPK_{(b)} + U_{50\%}$) enhanced seed N percentage. There are several possible explanations for the positive effects of bio-chemical fertilizers ($NPK_{(b)} + U_{50\%}$) on seed N Percentage. Chemical N and P fertilizers increase N import from vegetative parts to seeds, thereby increasing seed N content and protein percentage [29]. The effect of inoculation with indoleacetic acid-synthesizing and phosphate-solubilizing strains of *Azotobacter chroococcum* was found to be affirmative and significant on N uptake by wheat genotypes in greenhouse conditions [52]. It has been reported that some strains of *Azotobacter chroocuccum* increase N and P contents and their mobilization from roots to shoots in plants [53]. Therefore, with the combined use of bio-chemical fertilizers, in addition to reducing the amount of chemical fertilizers, it is possible to increase plant growth and thus consider environmental indices [18,54].

One of the main reasons for stimulating plant growth by biofertilizers is the secretion of various hormones as well as bacterial activity that leads to the solubilizing and availability of macro-elements (NPK) needed for plants [52]. The increase in N content and the stimulation of shoot growth of the plants are accompanied by the increase in P nutrition. Therefore, the effect of Pseudomonas on N content happens indirectly by improving the P status of the plants induced by bacterial symbiosis. In other words, Pseudomonas enhances N uptake by dissolving insoluble phosphate and increasing P availability for N fixing bacteria [55]. Kandeel et al. [56] on *Ocimum basilicum* reported that yield components and essential oil content and N, P, and K availability were increased through of inoculation *Azotobacter* sp. and *Azospirillum* sp. as biofertilizer.

#### 4.5.2. Seed P Content

It is apparent from our results that combined urea—biofertilizer ((NPK(b) + U50%) and P(b) + U50%)) had affirmative impact on seed P content. The effect of inoculation with indoleacetic acid-synthesizing and phosphate-solubilizing strains of *Azotobacter chroococcum* was found to be positive and significant on P uptake by wheat genotypes in greenhouse conditions [52]. It has been reported that in soils suffering from low P availability, the application of sulfur and its oxidizing bacteria can play a positive role in increasing P content in the shoots of yam bean (*Pachyrhizus erosus* L. *Urban*) [57]. Hassan et al. [58] found that biofertilizer inoculation resulted in more accessibility of nitrogen, phosphorus, and potassium. These interpretations confirm the findings of present research.

#### 4.5.3. Seed K Content

The effect of inoculation with indoleacetic acid-synthesizing and phosphate-solubilizing strains of *Azotobacter chroococcum* was found to be positive and significant on K uptake by wheat genotypes in greenhouse conditions [52]. In a study on the medicinal *Stevia rebaudiana Bert*, Das et al. [59] found that inoculation with *Azotobacter* sp. increased the accumulation of minerals, especially K, in plants, but the use of a mix of microorganisms (*Azotobacter* and phosphate-solubilizing bacteria) entailed further increase in K content due to their strong synergic relationships. It was established that K fertilization improved seed K content of lallemantia; the increase in seed K content with the increase in K treatment levels was attributed to the increase in the growth and development of the plant root system due to the application of K, which enhanced the area for K uptake [60]. These findings are in agreement with current research. In an experiment by Bedawi [61], it was

shown that the interactions of biofertilizers with different sources and amounts of nitrogen fertilizer increased vegetative growth and chemical content.

## 5. Conclusions

In general, it can be concluded that NPK (b) + U50% treatment had the most positive effect on the qualitative and quantitative traits. *Azotobacter* as a N-fixing bacteria, *Pantoea* and *Pseudomonas* as phosphate-solubilizing bacteria and *Bacillus* as potassium-solubilizing bacteria had key roles to ameliorate seed attributes. The treatment of NPK(b) + U50% enhanced seed yield, oil yield, seed NPK as macro –elements, and the unsaturated to saturated fatty-acids ratio. The results of our study showed that use of NPK (b) + U50% can be an effective practice for producing a large amount of high-quality seeds enriched with macro elements. According to the results, the integrated application of urea–biofertilizers can significantly reduce the use of chemical fertilizers and thus their detrimental environmental impacts, making progress towards sustainable farming objectives that lead to maintaining human health.

**Author Contributions:** S.M., writing—original draft; methodology and formal analysis, S.S.M., conceptualization, supervision and project administration, A.R., L.P., H.A.E.E. and R.Z.S. writing—review and editing, H.A.E.E., fund acquisition. All authors have read and agreed to the published version of the manuscript.

**Funding:** This research was funded by Allcosmos Industries Sdn. Bhd. Arif Efektif Sdn. Bhd., Malaysia with grant Ns. RJ130000.7609.4C187 and RJ130000.7344.4B200.

**Institutional Review Board Statement:** Not applicable.

**Informed Consent Statement:** Not applicable.

**Data Availability Statement:** All the data is available in the manuscript.

**Acknowledgments:** The authors are thankful to Allcosmos Industries Sdn. Bhd. Arif Efektif Sdn. Bhd., Malaysia for funding this work through grant Ns. RJ130000.7609.4C187 and RJ130000.7344.4B200 and the Faculty of agriculture, Urmia University, for their support.

**Conflicts of Interest:** All the authors confirm that there are no known conflicts of interest associated with this publication.

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
