# Peer review of "Bio-Chemical Fertilizer Improves the Oil Yield, Fatty Acid Compositions, and Macro-Nutrient Contents in Nigella sativa L."

_horticulturae, doi:10.3390/horticulturae7100345_

Round 1

Reviewer 1 Report

The article entitled “Bio-chemical fertilizer improves the oil yield, fatty acid compositions, and macro-nutrient contents in Nigella sativa L." presents the results of a study evaluated the effects of N biofertilizer (AzotoBarvar-1), P biofertilizer (PhosphoBarvar-2), and K biofertilizer (PotaBarvar-2) on qualitative and quantitative traits of Nigella sativa L. The use of preparations with microorganisms in agriculture has recently been quite popular and interesting. It requires further research. In this context, the research goal of the peer-reviewed article is interesting.

Detailed comments:

Please rephrase the abstract. Line 20 states that 10 treatments were tested, but it is not known what the treatments were. In line 22 (and below) the authors use the abbreviation NPKb + U50% - it should be explained what this means, especially b and U50%. In line 28 there is “the highest seed N content, seed P percentage, seed K content, and seed protein percentage” - why authors write content once, once percentage. The earlier part of the abstract does not support this statement “NPK(b) + U50% is the best treatment that reduces 29 chemical fertilizer by 50%”.

Keywords - in my opinion the keywords are wrongly chosen.

Line 59 – there is: “oleic acid, palmitic acid” - these acids are already listed on lines 53 and 54 - redundant repetition. There is no brief explanation of the role and function of these acids. Please consider adding a few sentences on this.

Line 84 – there should be potassium (no potash)

Line 107 – there is: “included urea and NPK biofertilizer and composed of urea  (U)” - it is unclear to me – please rephrase.

Line 172 - Why did the authors include a table with physicochemical characteristics of soil in the Statistical analysis section. This data should be in part “2.2. Soil preparation and Sowing”. Table 2 is also in the wrong section.

Table 3 - this table is unnecessary.

Line 193 - the studied plots did not cover 1 ha. The results should be given on the plot, not on the ha.

Figures need to be corrected. In this form, they don't look professional.

Figure 1 – in the section 2.12 it was not stated that the authors performed the Ducan test.

Figures 3-6 - the results of the statistical analysis are missing - please add this

The results in Figure 7 are strange. The nitrogen content in the seeds is up to 2%. There are very small differences between the individual research treatment - it is strange that these differences are statistically significant - please check the correctness of the statistical analysis.

In part Discussion, the authors presented a literature review, and this is not what the discussion is about. There is no comparison of the results obtained by the authors with the data from the literature.

Author Response

Response to comments of Reviewer 1 Round 1

  • Please rephrase the abstract. Line 20 states that 10 treatments were tested, but it is not known what the treatments were. In line 22 (and below) the authors use the abbreviation NPKb + U50% - it should be explained what this means, especially b and U50%. In line 28 there is “the highest seed N content, seed P percentage, seed K content, and seed protein percentage” - why authors write content once, once percentage. The earlier part of the abstract does not support this statement “NPK(b) + U50% is the best treatment that reduces 29 chemical fertilizer by 50%”.

Authors’ response: Re-phrased. Corrections done at Line 20, Line 22 and Line 29

  • Keywords - in my opinion the keywords are wrongly chosen.

Authors’ response: Revised

  • Line 59 – there is: “oleic acid, palmitic acid” - these acids are already listed on lines 53 and 54 - redundant repetition. There is no brief explanation of the role and function of these acids. Please consider adding a few sentences on this.

Authors’ response: Corrected. Lines 53 and 71

  • Line 84 – there should be potassium (no potash)

Authors’ response: Corrected. Line 100

  • Line 107 – there is: “included urea and NPK biofertilizer and composed of urea  (U)” - it is unclear to me – please rephrase.

Authors’ response: Corrected

  • Line 172 - Why did the authors include a table with physicochemical characteristics of soil in the Statistical analysis section. This data should be in part “2.2. Soil preparation and Sowing”. Table 2 is also in the wrong section.

Authors’ response: Corrected= lines 112 &124

  • Table 3 - this table is unnecessary.

Authors’ response: Tables of variance according to reviewer comments were deleted.

  • Line 193 - the studied plots did not cover 1 ha. The results should be given on the plot, not on the ha.

Authors’ response: It should be noted here that all field and even potted experimental designs should be expanded and stated per hectare, and it can be seen in various articles. Even the different amounts of fertilizer given per hectare are ultimately recommended to have a practical aspect.

  • Figures need to be corrected. In this form, they don't look professional.

Authors’ response: Corrected

  • Figure 1 – in the section 2.12 it was not stated that the authors performed the Ducan test.

Authors’ response: Corrected

  • Figures 3-6 - the results of the statistical analysis are missing - please add this

Authors’ response: Fatty acids did not have replications, so statistical analysis could not be performed. In fact we didn’t analyze the samples with 3 replications. However, before analyzing, the 3 replications were mixed. Many studies due to accuracy of GC –mass instrument analyze fatty acids without replications such as the below published article in Q1 journal:

Mohammadghasemi, V., Moghaddam, S. S., Rahimi, A., Pourakbar, L., & Popović-Djordjević, J. (2020). The Effect of Winter Sowing, Chemical, and Nano-Fertilizer Sources on Oil Content and Fatty Acids of Dragon’s Head (Lallemantia iberica Fischer & CA Meyrefeer). Journal of Plant Growth Regulation, 1-14 .

  • The results in Figure 7 are strange. The nitrogen content in the seeds is up to 2%. There are very small differences between the individual research treatment - it is strange that these differences are statistically significant - please check the correctness of the statistical analysis.

Authors’ response: Checked statistical analysis. It is correct. It should be noted that significant differences are not related to small or large numbers and number differences.

It is related to Mean Square of Error (variance of error). The smaller Mean Square of Error, lead to the more significant the differences.

  • In part Discussion, the authors presented a literature review, and this is not what the discussion is about. There is no comparison of the results obtained by the authors with the data from the literature.

Authors’ response: Corrected

Reviewer 2 Report

Please consider the following notes:

The description in line 96 is not to clear. Is it meant that a plot is 3 m × 2.5 m, i.e. 7.5 m2? And therefore 200 plants per plot. There are 3 x 10 plots in total. Please describe the plot size correctly.

In Material and Methods, a table is required to show how much N, P and K was added in each treatment with the fertilizer application.

Line 174 indicates that the PLSD (protected least significant difference-Fisher) test is used for the mean comparison. In fact, according to the figures, the Duncan test was used. The tests are similar, but not the same. Since all treatments were compared with each other, these tests are not suitable; instead, the Tukey test or the Scheffe test should be used. Please correct this.

The calculation of the protein % is determined by a factor of 5.7* N % (chapter 2.8 and 2.9). A statistical evaluation (correlation as well as ANOVA and mean value comparisons) is thus not meaningful, since the results are preprogrammed. It is all the more surprising, that the correlation coefficient (Tab. 5) is only 0.95. This is obviously due to the fact that N % was not used for the correlation. Only the statistical data for N or protein should be presented. The other value can be discussed in the text.

The acronyms for the treatments, as used later in the figures and abstract, are to be introduced in the chapter Material and Methods part under 2.4.
According to Figure 2, all figures should have the acronyms used on the x-axis. The detailed listing can thus be omitted in the figure caption.

Notes on the individual figures:
Figure 1: the two characteristics: seed yield and oil yield can be plotted in a bar chart with two data sets, each containing 10 treatments (compare Figure 2). The letters must be different for the characteristics, for example small letters and capital letters.
Figures 3, 4, and 5: No comparison of means is made; the data for the treatments are presented. Are the differences not significant or what is the cause? In these figures, both data sets should be plotted as a bar chart, not one data set as a column and the other as a line. This is nonsensical primarily because these are categories.

Figures 7 and 8: please consider the comments on figure 1.

In the discussion (Chapter 4), literature results are presented in detail, but little reference is made to the author's own results as to whether they confirm the literature results or whether the literature results confirm the results presented here. This needs to be supplemented.

The conclusions are formulated too simply. The importance of the individual nutrients N, P and K should be mentioned here. No significant differences were shown for the characteristics palmitic, stearic and linoleic acids.

Author Response

Response to comments of Reviewer 2 Round 1

  • The description in line 96 is not to clear. Is it meant that a plot is 3 m × 2.5 m, i.e. 7.5 m2? And therefore 200 plants per plot. There are 3 x 10 plots in total. Please describe the plot size correctly.

Authors’ response: The study was carried out on the basis of a randomized complete block design with 10 treatments and three replications (30 plots). Each plot was 3 × 2.5 m2

  • In Material and Methods, a table is required to show how much N, P and K was added in each treatment with the fertilizer application.

Authors’ response: Bacterial NPK are inoculated with seeds. The amount is not important. The procedure described in M&M.

  • Line 174 indicates that the PLSD (protected least significant difference-Fisher) test is used for the mean comparison. In fact, according to the figures, the Duncan test was used. The tests are similar, but not the same. Since all treatments were compared with each other, these tests are not suitable; instead, the Tukey test or the Scheffe test should be used. Please correct this.

Authors’ response: Corrected to PLSD. Also, considering that F was significant and we wanted to identify minor changes in the means, so we used this test, in which even small changes are significant specially in fatty acids.

  • The calculation of the protein % is determined by a factor of 5.7* N % (chapter 2.8 and 2.9). A statistical evaluation (correlation as well as ANOVA and mean value comparisons) is thus not meaningful, since the results are preprogrammed. It is all the more surprising, that the correlation coefficient (Tab. 5) is only 0.95. This is obviously due to the fact that N % was not used for the correlation. Only the statistical data for N or protein should be presented. The other value can be discussed in the text.

Authors’ response: We kept N, and Protein was deleted. Correlation table was removed.

  • The acronyms for the treatments, as used later in the figures and abstract, are to be introduced in the chapter Material and Methods part under 2.4.
    According to Figure 2, all figures should have the acronyms used on the x-axis. The detailed listing can thus be omitted in the figure caption.

Authors’ response: Corrected=line 132

  • Figure 1: the two characteristics: seed yield and oil yield can be plotted in a bar chart with two data sets, each containing 10 treatments (compare Figure 2). The letters must be different for the characteristics, for example small letters and capital letters.

Authors’ response: It should be mentioned that seed yield and oil yield are two separate graphs with separate statistical analysis.

  • Figures 3, 4, and 5: No comparison of means is made; the data for the treatments are presented. Are the differences not significant or what is the cause? In these figures, both data sets should be plotted as a bar chart, not one data set as a column and the other as a line. This is nonsensical primarily because these are categories.

Authors’ response: Fatty acids did not have replications, so statistical analysis could not be performed. In fact we didn’t analyze the samples with 3 replications. However, before analyzing, the 3 replications were mixed. Many studies due to accuracy of GC –mass instrument analyze fatty acids without replications such as the below published article in Q1 journal:

Mohammadghasemi, V., Moghaddam, S. S., Rahimi, A., Pourakbar, L., & Popović-Djordjević, J. (2020). The Effect of Winter Sowing, Chemical, and Nano-Fertilizer Sources on Oil Content and Fatty Acids of Dragon’s Head (Lallemantia iberica Fischer & CA Meyrefeer). Journal of Plant Growth Regulation, 1-14

  • Figures 7 and 8: please consider the comments on figure 1.

Authors’ response: Corrected

  • In the discussion (Chapter 4), literature results are presented in detail, but little reference is made to the author's own results as to whether they confirm the literature results or whether the literature results confirm the results presented here. This needs to be supplemented.

Authors’ response: Corrected

  • The conclusions are formulated too simply. The importance of the individual nutrients N, P and K should be mentioned here. No significant differences were shown for the characteristics palmitic, stearic and linoleic acids.

Authors’ response: Corrected=line 427

Reviewer 3 Report

The paper ris of interest and well documented.

In all the manuscript, the units after the decimal point must be checked.

P2, l78 : « simulator » or stimulator ?

P3, l130 : check « inhibition times » ; probably to be replaced by « retention times ».

P3, l132 : check the word « decomposition »

P3, l134 : clarify the sentence : « The flame detector (FID) can approach zero µm.

P6, figure 2 : the titles of the axes are difficult to read.

P6, l219 : « 0.65 » must be replaced by « 0.65% ».

P7, figures 3, 4, 5, 6 : the titles and legends of the axes are difficult to read.

P8, 240-241 : « The results of analysis of variance for the effect of chemical urea and biofertilizers on NPK 240 accumulation in seeds and leaves (Table 4). » : This sentence is missing a verb.

P12, l405 : replace « linoleic » by « linolenic ».

Author Response

Response to Comments of Reviewer 3 Round 1

The paper ris of interest and well documented.

P1. In all the manuscript, the units after the decimal point must be checked.

Authors’ response: Corrected

P2, l78 : « simulator » or stimulator ?

Authors’ response: Corrected. Line 93

P3, l130 : check « inhibition times » ; probably to be replaced by « retention times ».

Authors’ response: Corrected. Line 156

P3, l132 : check the word « decomposition »

Authors’ response: Deleted

P3, l134 : clarify the sentence : « The flame detector (FID) can approach zero µm.

Authors’ response: The flame ionization detector (FID) is a standard instrument used in industry for measuring, but its response is either poor or nil to compounds. It is famous method for chemists.

P6, figure 2 : the titles of the axes are difficult to read.

Authors’ response: Corrected. Line No.  232

P6, l219 : « 0.65 » must be replaced by « 0.65% ».

Authors’ response: Corrected

P7, figures 3, 4, 5, 6 : the titles and legends of the axes are difficult to read.

Authors’ response:

P8, 240-241 : « The results of analysis of variance for the effect of chemical urea and biofertilizers on NPK 240 accumulation in seeds and leaves (Table 4). » : This sentence is missing a verb.

Authors’ response: According to reviewers comment (comment 8, reviewer1 as unnecessary table), Table of variance deleted.

P12, l405 : replace « linoleic » by « linolenic ».

Authors’ response: Corrected. Line No,  427

Round 2

Reviewer 1 Report

I appreciate the author' efforts on this manuscript, which indeed improve the quality of this manuscript. Particularly,  the authors added missing information, updated data, improved Figures. Thus, I satisfy the authors' respondence and the revision.